# The influence of personal care products on ozone-skin surface chemistry

**Glenn Morrison**[1]*, **Azin Eftekhari**[1], **Aixing Fan**[2], **Francesca Majluf**[3], **Jordan E. Krechmer**[3]

**1** Department of Environmental Sciences and Engineering, Gillings School of Global Public Health, The University of North Carolina at Chapel Hill, Chapel Hill, North Carolina, United States of America, **2** Colgate-Palmolive Co., Piscataway, NJ, United States of America, **3** Aerodyne Research Inc., Billerica, Massachusetts, United States of America

\* glenn.morrison@unc.edu

## Abstract

Personal care products are increasingly being marketed to protect skin from the potentially harmful effects of air pollution. Here, we experimentally measure ozone deposition rates to skin and the generation rates and yields of oxidized products from bare skin and skin coated with various lotion formulations. Lotions reduced the ozone flux to the skin surface by 12% to 25%; this may be due to dilution of reactive skin lipids with inert lotion compounds or by reducing ozone diffusivity within the resulting mixture. The yields of volatile squalene oxidation products were 25% to 70% lower for a commercial sunscreen and for a base lotion with an added polymer or with antioxidants. Lower yields are likely due to competitive reactions of ozone with lotion ingredients including some ingredients that are not intended to be ozone sinks. The dynamics of the emissions of squalene ozonation product 6 methyl-2-heptenone (6MHO) suggest that lotions can dramatically reduce the solubility of products in the skin film. While some lotions appear to reduce the rate of oxidation of squalene by ozone, this evidence does not yet demonstrate that the lotions reduce the impact of air pollution on skin health.

## Introduction

Evidence of an adverse effect of air pollution on skin health has been growing for the past 20 years [1]. Epidemiological studies report observations of relationships between air pollutants and atopic dermatitis [2], skin aging [3], acne [4] and multiple cause-specific skin conditions [5]. Plausible mechanisms include generation of free radicals and oxidation products that induce an inflammatory cascade and disruption of the barrier function of the skin [6]. For example, squalene hydroperoxide (SqOOH) is suspected of participating in the pathogenesis of acne [7]. The lipid peroxidation product 4-hydroxy-2-nonenal (4-HNE) is a result of oxidative stress and has been associated with skin aging [8]. Both molecules have been observed as products of skin exposure to ozone [8].

Ozone chemistry with skin has been studied widely over the past decade after the discovery that occupants of buildings can be an important reactive sink for ozone [9]. Ozone reacts

**Funding:** This research was financially supported by Colgate-Palmolive, Inc. and the Alfred P. Sloan Foundation through the Modelling Consortium for

Chemistry of Indoor Environments (MOCCIE 2, G-2019-12306).

**Competing interests:** The authors have declared that no competing interests exist.

readily with double bonds present in squalene, fatty acids and triglycerides, forming products including volatile carbonyls, acids, ozonides and others [9–16]. Volatile products of ozone-skin surface lipids have been observed in the air of homes [17], offices [10], classrooms [18] and stadiums [19]. In our previous study [20], we quantified the yields of 40 products of ozone chemistry with the bare forearm skin of 20 adults. We observed that yields of squalene ozonation products varied substantially across the population of human subjects. The summed yields of all products ranged from 0.33 to 0.93. Factor analysis indicated that the variability of the bare-skin chemistry was due to the relative abundance of fresh endogenous skin surface lipids, oxidized or "aged" lipids and exogenously sourced lipids (e.g. from cooking oils and personal care products). This last item indicates that skin chemistry can be influenced by contact with the multitude of products we encounter daily. Intentional application of personal care products, such as lotions, could also alter skin chemistry, even if only by dilution of native skin surface lipids.

The goal of this research was to quantitatively assess how much lotions influence the chemistry of ozone with the skin surface. Some of the lotions tested had additives intended specifically to either react with ozone or block it. To make statistically significant comparisons between bare skin and lotions of different compositions, we recruited 20 adults for direct measurements on the skin of the forearm. We had two specific objectives: 1) quantify and compare the ozone flux (and ozone deposition velocity) to bare skin and skin with applied lotions and 2) quantify and compare volatile skin-surface ozonation product yields. We met these objectives and show how lotions alter the yield and dynamic release of products from ozone reactions with skin surface lipids. To our knowledge, this research is the first to assess the influence of lotions on ozone-skin lipid chemistry, which has implications for both skin health and indoor air quality.

## Methods

This project and its methods have been described in detail in Morrison et al. [20]. Therefore, the following description of methods is abbreviated but highlights method descriptions more specific to this research.

### Participants

A convenience sample of 21 volunteers participated in this study; because of incomplete data collection, results from one participant are excluded from the analysis. Therefore, we include results for a 20 participants (17 female, 3 male; age 20–61 y). Participants were asked to shower on the morning of the study and refrain from use of personal care products that day. For each participant, the experimental procedure took about 1.5 h which included the completion of a questionnaire, application of lotions and measurements at six locations on the arms. Participants were given a $50 gift card as compensation. The study plan (#18–3126) was approved by the University of North Carolina, Chapel Hill Institutional Research Board. Recruitment documents and intake questionnaires are shown in section S1 of S1 File.

### Materials

The study sponsor (Colgate-Palmolive, Inc.) provided five lotion formulations. Available details of the formulations are shown in Table 1. Lotion 1 is commercial moisturizing lotion that was used as a base for lotions 2–4. Lotions 2,3 and 4 have a polymer, a mixture of waxy lipids and an antioxidant blend added to lotion 1, respectively. Some ingredients and amounts added are proprietary and all are cosmetic grade. Lotion 5 is a commercial sunscreen with no other additives.

**Table 1. Lotion ingredients.**

| Number | Lotion | Ingredients |
|---|---|---|
| 1 | Base lotion: Sanex body lotion (market product) | Deionized water, Propylene glycol, Sodium benzoate, Lactic acid–food grade, Caprylyl glycol, Glycerin, White mineral oil, Glyceryl monostearate, Stearyl alcohol, Silicone 350CS, PEG 100 stearate, Cetyl alcohol |
| 2 | Base lotion plus waxy lipids | Additive: proprietary mixture of waxy lipids |
| 3 | Base lotion plus copolymer | Additive: proprietary copolymer |
| 4 | Base lotion plus antioxidants | Additive: proprietary mixture of antioxidants |
| 5 | Elta MD UV broad-spectrum shield SPF 45 (market product) | Active: Zinc Oxide 9.0%, Octinoxate 7.5%. Inactive: Purified Water, Isopropyl Palmitate, Octyl Stearate, Cetearyl Alcohol, Polysorbate 60, Oleth-3 Phosphate, Phenoxyethanol, Cetearyl Glucoside, Hydroxyethyl Acrylate/Sodium Acryloyldimethyl Taurate Copolymer, Polyisobutene, Polyether-1. Butylene Glycol, PEG-7 Trimethylolpropane Coconut Ether, Tocopheryl Acetate, Citric Acid, Iodopropynyl Butylcarbamate, Triethoxycaprylylsilane. |

## Instruments and experimental apparatus

The overall experimental system is described in detail in section S2 of S1 File and is shown in S1-S6 Figs of S1 File. Approximately 0.17 L/min of ultrapure air was humidified (approximately 60% RH) and directed through an ozone generator before flowing through a skin-surface sampling chamber and finally to air composition analyzers. Three Teflon-body valves were used to direct the air to the ozone generator, or to bypass it, and to the surface sampler, or to bypass it. The skin-surface samplers were made from solid perfluoroalkoxy (PFA) disks which were machined to create flow-through flux chambers that exposed 2.0 cm$^2$ of skin area (see diagrams in S2 Fig of S1 File). During operation, a 0.25mm thick gasket of medical grade silicone separated the surface sampler from the skin and created a seal. With air flowing, the samplers operate at a slight overpressure, preventing room air from entering the sampler. Samplers were cleaned with lab-grade detergent and methanol and dried in at 60˚C overnight before use with participants.

The mixing ratio of ozone was measured using a UV-photometric ozone analyzer (Monitor Labs ML9811 Photometric Ozone Analyzer with analog output logged externally by DATAQ module and computer; limit of detection 1 ppb, uncertainty under experimental conditions with dilution ~4 ppb) was used to monitor ozone. When ozone was flowing into the cell over skin, the outlet mixing ratio ranged from 100 to 120 ppb. Volatile products of ozone chemistry were quantified using an Aerodyne/Tofwerk Vocus proton-transfer reaction time-of-flight mass spectrometer (Vocus PTR-TOF-MS) [21]. The PTR-TOF-MS calibration procedure is detailed in sections S3 and S4 of S1 File. High-resolution PTR-TOF-MS can provide elemental formulae of ions, but not molecular structures. We rely in prior studies of skin lipid ozonation to assign ions to unique species. For species that appear to be due to ozone chemistry with lotion components, we rely on the reported ingredient list to help understand the chemistry and suggest possible products. Mixing ratios of two compounds, GA and 6-MHO, were corrected based on observed fragmentation ratios during pure-compound "sniff" tests (see section S4 of S1 File) and further corrected based on observed uptake and oxidation taking place in the sample line (see Morrison et al., 2021). The mixing ratios were then normalized by the sampler outlet ozone mixing ratio when the sampler was positioned over a clean PFA surface minus the mixing ratio when the sampler was positioned over the skin (ppb; limit of quantification of difference ~2 ppb, uncertainty in difference ~5%); this resulted in a surface-specific yield. Note that the calibration factors for these compounds have a higher uncertainty since a calibration standard with the majority of oxidation products was not available at the time of the experiment. We do not report ozone-initiated yields of acetone due to high background levels that may have come from lotions themselves.

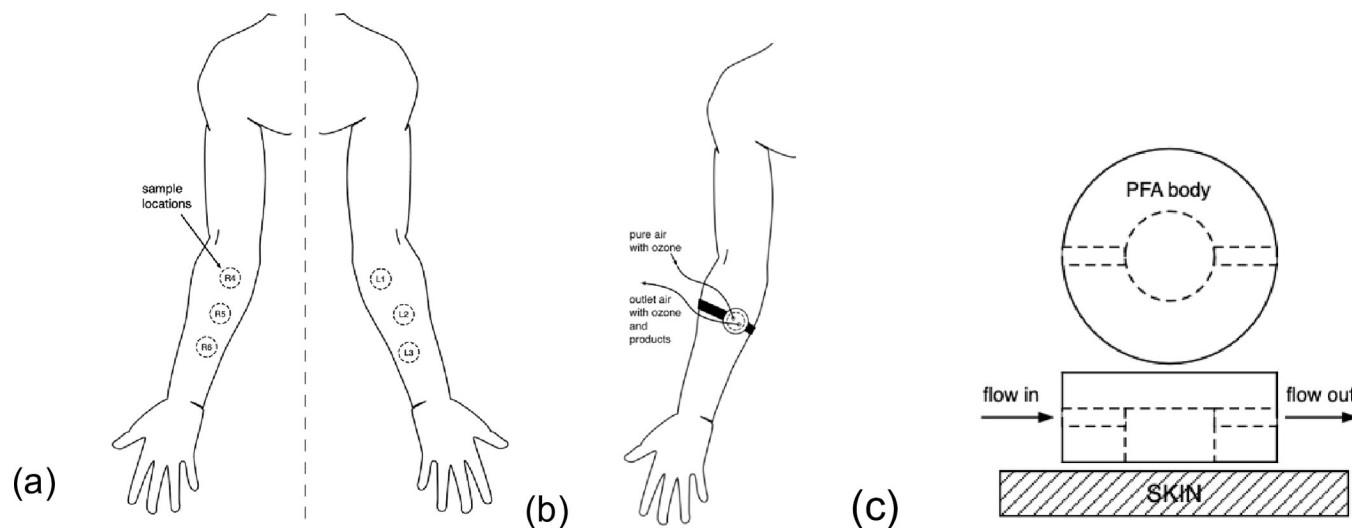

**Fig 1. Diagrams of sampling locations and flow cell.** a) Sample locations on forearm; b) attached flow cell; c) flow cell plan view and cross-section (gauze not shown).

## Procedure

The first step with each participant was to mark out three square 10 cm$^2$ locations for testing on the inner forearm of each arm; these approximate locations are shown in Fig 1. Location L1 (left arm 1) was set aside as the bare-skin location for all participants. Lotions 1–5 were then randomized to the other locations. Prior to initiating an experiment, 20 mg of the chosen lotion was applied to location L2 (left arm 2) and allowed to dry for 10 minutes. The lotion was applied with a thin plastic dowel, rubbing lightly across the entire area vertically and horizontally several times, to spread the lotion evenly based on a visual inspection. At this time, the surface sampler was placed on location L1 (bare skin) and a 10 min ozone exposure sequence was initiated (see below) and the next lotion was applied to location L3. After the exposure sequence on L1, the sampler was moved to L2. Note that at this time, the lotion that had been applied to L2 has dried for 20 minutes. Then an ozone exposure sequence was initiated for the surface sampler located at L1 and the next lotion was applied to location R1. This procedure was continued until all locations were coated with lotion, allowed to dry for 20 minutes and subject to an ozone exposure sequence. Lotion application was consistent with an average of 2.04±0.12 mg/cm$^2$ over all lotion applied and a minimum and maximum applied of 1.75 and 2.35 respectively. Results by lotion are shown in S5 Fig of S1 File.

The ozone exposure sequence for each arm location took 10 minutes, starting with 1 minute of ozone-free air passing through the sampler to the instruments (no-ozone background measurement). This was followed by a 2-minute period of sampler bypass to stabilize the upstream ozone mixing ratio without delivering ozone to the surface sampler. At this 3-minute mark, ozone was allowed to flow through the skin-surface sampler for 5 min while instruments analyzed the outlet composition. Then the ozone generator was bypassed, shutting off ozone supply to the surface sampler and allowing the remaining ozone-initiated reaction product mixing ratios to decay. After 2 min (total 10 min) the sequence was ended, and the surface sampler moved to the next arm location. A more detailed start-to-finish experimental sequence is provided in Section S2 of S1 File.

Two experiments were performed after reviewing preliminary results from the participant study. In the first, a small amount of lotion was applied to a 2.5 cm diameter cellulose filter and

allowed to air dry for 20 minutes. These were shipped to Aerodyne and "sniff" tested with the PTR-TOF-MS to observe ions resulting from primary emissions. In the second, one of the known components of lotion 5, pure octinoxate, was exposed to ozone to identify ozonation products unique to this additive.

## Data analyses

The ratio of the outlet ozone mixing ratio, $C_{O3,outlet}$, to the blank outlet mixing ratio, $C_{O3,blank}$, is used to assess variability in ozone consumption among participants' skin. The blank outlet is determined by placing the flux cell on back of a second clean PFA flux cell to measure outlet ozone mixing ratio prior to placing the device on the skin. By using $C_{O3,blank}$ we account for small ozone losses on inner surface of the flux cell itself.

$$R_{O3} = \frac{C_{O3,outlet}}{C_{O3,blank}} \tag{1}$$

This ratio is used to quantify the ozone deposition velocity, $v_d$,

$$v_d = \frac{Q_{cell}}{A_{skin}} (R_{O3} - 1) \tag{2}$$

Product yield for species $i$, $Y_i$, is defined as,

$$Y_i = \frac{C_{i,O3} - C_{i,background}}{C_{O3,blank} - C_{O3,skin}} \tag{3}$$

where $C_{i,O3}$ is the mixing ratio of species $i$ in the presence of ozone, averaged during the last 10 seconds of sequence step 3 (S8 Fig of S1 File; 07:50–08:00). $C_{i,background}$ is the mixing ratio of species $i$ in the absence of ozone during the last 10 seconds of sequence step 1 (S6 Fig of S1 File; 01:50–02:00).

## Statistical analysis

For each participant and lotion, results are reported with uncertainty determined based on replicate measures across all participants. We applied a paired sample t-test using Matlab 2020b on ozone uptake parameters (i.e. deposition velocity) and product yields. The results from lotion experiments are considered significantly different when the p-value is $<0.05$.

## Results and discussion

### Ozone consumption

Shown in Fig 2A and 2B are the ozone ratio $R_{O3}$ and the deposition velocity, $v_d$. For this configuration of flux cell and flowrate, about 20% of ozone delivered reacts with bare skin. All lotions reduced ozone uptake somewhat with only 15–18% of ozone is removed at lotion covered skin. The deposition velocity for lotion covered skin is about 12% to 25% lower than for bare skin (p≤0.05; * in the figure). These reductions may simply be the result of diluting high-reactivity skin lipids with low-reactivity lotion ingredients. Although the application of lotion 1 to the skin dilutes skin lipids by approximately a factor of 5 (see discussion in Section S7 of S1 File), this does not result in a proportional reduction in the deposition velocity. This is probably because ozone uptake remains gas-side mass-transport limited: even a factor of 5 reduction in surface reaction probability does not reduce the deposition velocity proportionally. See Section S7 of S1 File for a more detailed explanation. Additives to lotions 2 and 3 did not appear to influence the ozone deposition velocity relative to base lotion 1. Lotion 4 had a significantly

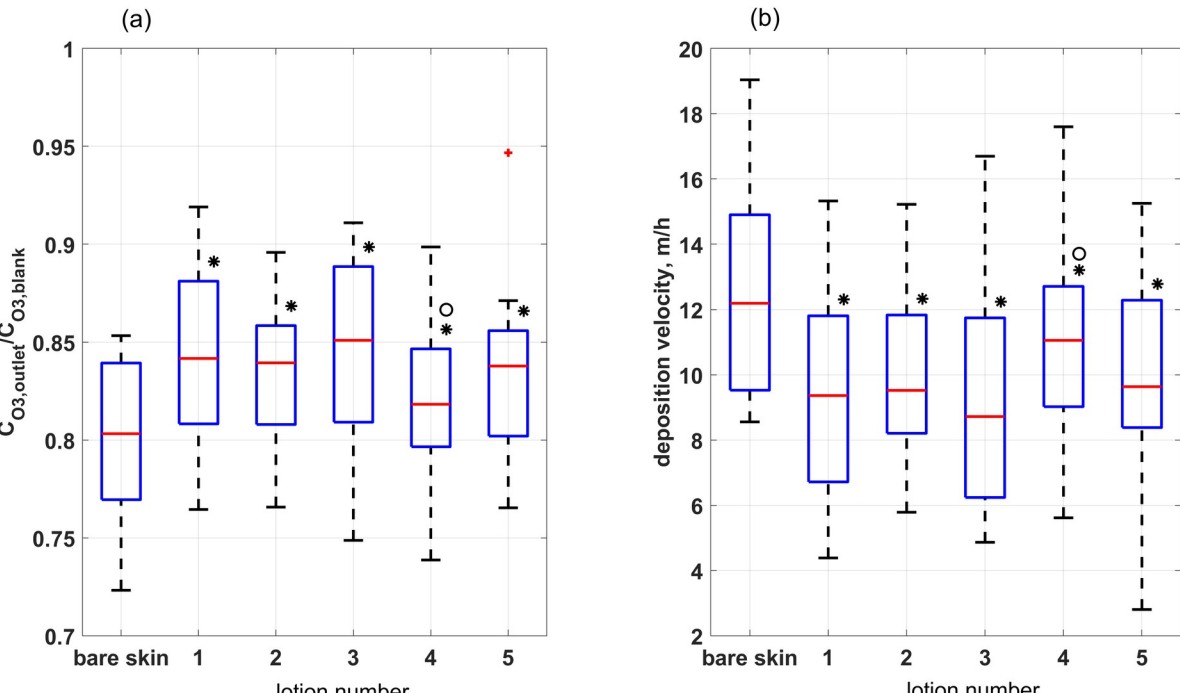

**Fig 2. Ozone mixing ratios and deposition velocities.** a) ratio of outlet to inlet ozone mixing ratios; b) deposition velocity. The symbol (*) indicate that results are significantly different from bare skin for $p \leq 0.05$. They symbol (°) indicates that results are significantly different from lotion 1.

higher deposition velocity than lotion 1 ($p<0.05$ noted by the symbol in the figure), indicating that the additives (probably the antioxidants) in lotion 4 slightly increased the ozone reactivity of the base lotion.

## Primary emissions from lotions

Mixing ratios of major ions and compound assignments, associated with primary volatile emissions (prior to introduction of ozone) from lotions, after application to skin, are shown in Table 2. All lotions emitted substantial amounts of propylene glycol (a listed ingredient), acetone and propionic acid. Lotions 1–4 also emitted octane glycol (a listed ingredient), benzoic acid and possibly octanoic acid. The flux cell outlet mixing ratios are very similar across lotions 1–4; this is not surprising given that they each are composed primarily of base lotion 1. No ions associated with primary emissions of additives to lotions 2–4 were observed. Lotion 5 emitted much more phenol than lotions 1–4. Lotion 5 also emitted phenoxyethanol (a listed ingredient) and possibly 4-methoxy benzaldehyde. We also observed a large ion signal that may be associated with the dehydration of phenoxyethanol in the VOCUS. We assign $C_8H_8O_2$ to 4-methoxybenzaldyde because this is a possible oxidation product of octinoxate, a UV inhibitor added to lotion 5 at 7.5% (See *Yields of other significant ions* section). Ions associated with primary emissions from lotions that had been applied to cellulose filters matched those shown in Table 2.

## Ozone-skin lipid product yields

Ozone initiated product yields of major products (6MHO, GA and decanal) associated with ozonation of skin lipids are shown in Fig 3A. As reported for bare skin (Morrison et al., 2021),

**Table 2. Select ions associated with primary emissions from lotions.** Mixing ratios (ppb) shown have baseline (bare skin) values subtracted (by subject), then averaged.

| Ion | Compound assignment | Lotion # | | | | |
|---|---|---|---|---|---|---|
| | | Mean mixing ratio (ppb) | | | | |
| | | **1** | **2** | **3** | **4** | **5** |
| $C_2H_5O^+$ | acetaldehyde | 37 | 37 | 42 | 34 | 8.2 |
| $C_3H_7O^+$ | acetone | 959 | 954 | 1080 | 879 | 113 |
| $C_3H_7O_2^+$ | propionic acid | 117 | 117 | 133 | 108 | 11.4 |
| $C_3H_9O_2^+$ | propylene glycol | 95 | 96 | 108 | 88 | 9.2 |
| $C_6H_7O^+$ | phenol | 12 | 11.9 | 9.7 | 8.6 | 292 |
| $C_8H_9O^+$ | dehydration fragment of phenoxyethanol | 8.5 | 8.8 | 6.9 | 5.4 | 332 |
| $C_7H_7O_2^+$ | benzoic acid | 14 | 13 | 12 | 13 | 8.4 |
| $C_8H_9O_2^+$ | 4-methyl benzaldehyde | 0.20 | 0.23 | 0.17 | 0.15 | 4.0 |
| $C_8H_{11}O_2^+$ | phenoxyethanol | 3.1 | 3.1 | 2.5 | 1.9 | 118 |
| $C_8H_{17}O_2^+$ | octanoic acid | 1.7 | 1.5 | 1.6 | 1.6 | 0.65 |
| $C_8H_{19}O_2^+$ | caprylyl glycol | 0.39 | 0.34 | 0.36 | 0.35 | 0.11 |

the yields for 20 subjects cover a broad range. For example, the 6MHO yield ranges from less than 0.1 to greater than 0.4. Although not obvious from Fig 3A, yields from lotions 2, 4 and 5 are significantly lower than for bare skin. To make this easier to visualize, we divided the yield associated with each lotion and participant by the bare skin yields for that participant; this normalization, shown in Fig 3B, more explicitly shows the differences among lotions in a pairwise fashion for each subject.

Yields of skin-lipid related volatile ozonation products (GA, 6MHO and decanal) all are lower for lotions 2, 4 and 5. As a reminder, yields are assumed to be independent of the flux or deposition velocity. Also, the overlying ozone concentration is similar among the experiments. Therefore, ozone is being consumed at a rate slightly lower than bare skin, but lipid oxidation products are released from the skin at a much more reduced rate from lotions 2, 4 and 5 relative to bare skin. For the same overlying ozone concentration and mass transfer conditions, it is possible to combine the deposition velocity and yield results to show that the emission rates of GA, for example, would be lower by 54%, 53% and 73% for lotions 2, 4 and 5 respectively. Also interesting is the narrowing of the distribution of normalized yields for lotions 2, 4 and 5. In other words, some lotions appear to reduce the variability in lipid oxidation rates or product release rates. One possible mechanism is that the composition and properties of the resulting mixture are more uniform, such as water content, viscosity, and pH.

The yield reduction by lotions may be due to lotion additives that are competing for ozone and reducing the rate of lipid ozonolysis. Lotion 2 includes a blend of proprietary waxy lipids, some of which may be unsaturated and therefore react readily with ozone. Lotion 4 includes a blend of proprietary cosmetic grade antioxidants. Some antioxidants used commonly in cosmetics can react directly with ozone. For example Vitamin C (ascorbic acid), commonly added to cosmetics, might compete for ozone since it has a meaningful rate of reaction with ozone [22, 23]. Others may not react directly with ozone but instead react with free radicals that are products of ozone chemistry.

Of the inactive ingredients present in lotion 5, only oleth-3-phosphate is unsaturated and likely to react with ozone. Many of the ingredients high on the list are derivatives of saturated fatty acids such as isopropyl palmitate and octyl stearate which do not readily react with ozone. The active ingredient octinoxate, added as an ultraviolet light inhibitor (sun protection ingredient) has a double bond that appears not to be sterically or otherwise hindered from reaction with ozone. In fact, upon exposure to ozone, compounds are emitted that are consistent with

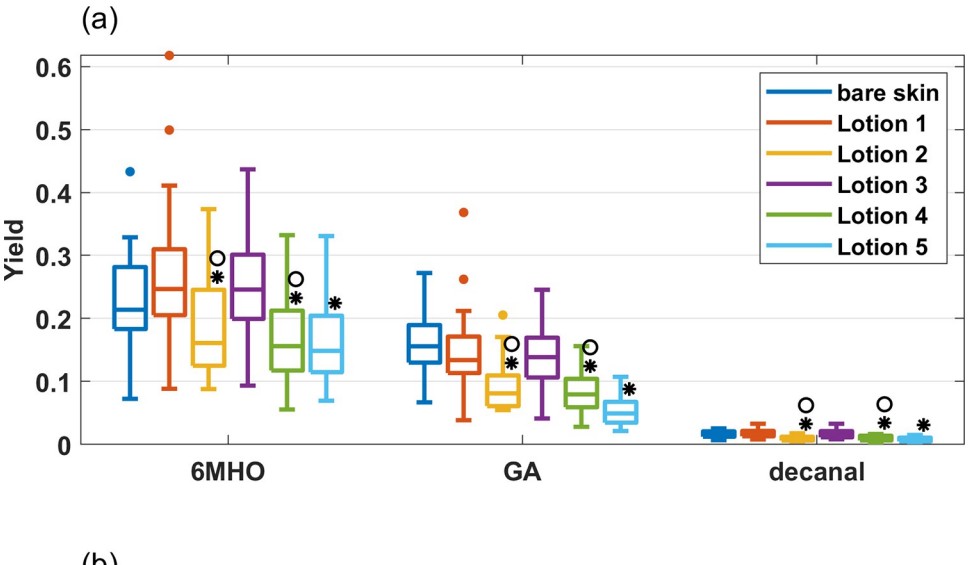

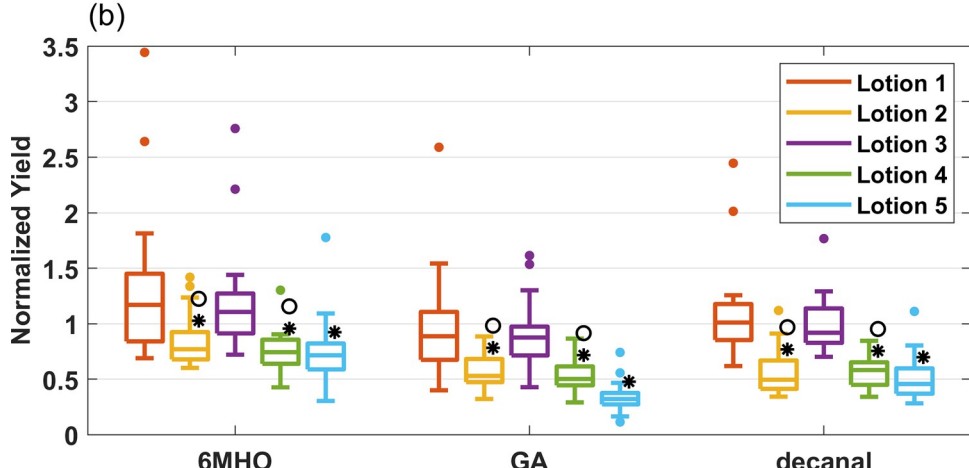

**Fig 3.** a) Yields of 6MHO, GA and decanal for bare skin, and lotions 1 through 5; b) Yields of 6MHO, GA and decanal normalized by bare skin for lotions 1 through 5. The central horizontal line is the median for 20 subjects. The upper and lower extents of the boxes represent the 25th and 75th percentiles and whiskers represent maximum and minimum, excluding outliers. Outliers are defined as values that are more than 1.5 times the interquartile range away from the top and bottom of a box. Asterisks indicate results that are significantly different from bare skin, specifically where $p \leq 0.05$. The symbol ˚ indicates lotions (within the lotion 2–4 group) that had a significantly different yield than lotion 1.

probable products of that reaction, as discussed the following section. Therefore, it is plausible that lower yields of 6MHO, GA and decanal for lotion 5 are due to competition for ozone with octinoxate.

## Yields of other significant ions

We observed that the lotions not only influenced yields of expected products of ozone reactions with skin lipids. We also observed products formed that are possibly due to reactions of ozone with lotion components. Some species are tentatively identified while others are characterized only be their ion formula. To provide a visual guide to the differences in the yield patterns among lotions, we collected ions into groups organized by carbon number and/or specific compound and likely ion fragment. For example, all ions with formulae $C_8H_xO_yH^+$

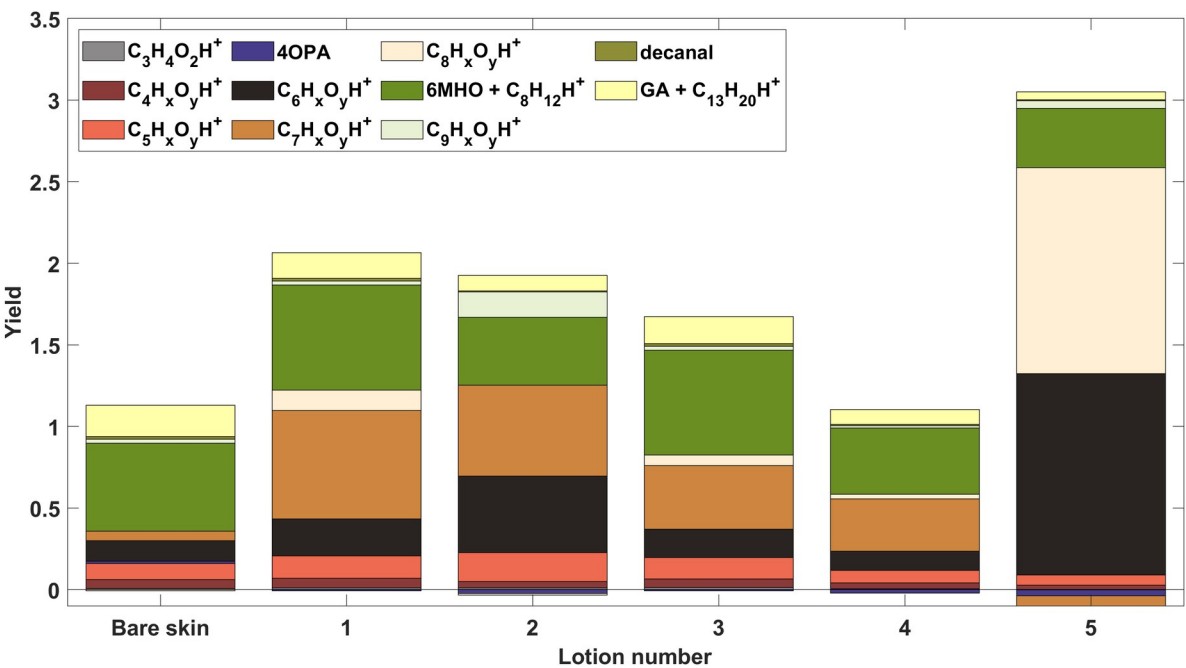

**Fig 4. Product yields.** Yields of individual species as well as ions grouped by number of carbons.

were placed in one group except for GA and its probable fragment ion, $C_8H_{13}^+$ which are plotted separately. Similarly, the groups $C_5H_xO_yH^+$, $C_6H_xO_yH^+$, $C_7H_xO_yH^+$, and $C_9H_xO_yH^+$ collect ions by carbon number. The results are shown as a stacked bar chart in Fig 4. Except for some $C_7H_xO_yH^+$ group ions, which exhibit positive and negative yields, we removed any species exhibiting negative yields that we determined were related to a decay in the rates of primary emissions and were therefore not related to ozone chemistry.

Lotions 1–4, which are based on the same lotion (1) but with different additives, exhibit different yield patterns. Lotions 2–4 all exhibit increases in the yield of $C_7$ ions dominated by $C_7H_7O_2^+$ and $C_7H_5O^+$ which could represent benzoic acid and its dehydration product, respectively. Lotion 2 had distinctive increases in yields of groups $C_6$ and $C_9$. The additive ingredients of lotion 2, a proprietary mixture of waxy lipids could explain these groups, if the lipids themselves are unsaturated. There was a large relative increase in the $C_9$ group for lotion 2 associated with nonenal (or an isomer) and its possible dehydration product $C_9H_{15}^+$. Nonenal can be generated from the reaction of ozone with fats containing two or more double bonds such as linoleic acid or a similar. If linoleic acid is present, ozone attack at the omega-6 double-bond could also explain the increase in $C_6$ alkyl fragments observed from lotion 2.

Lotion 5 had a very different pattern of yields from lotions 1–4. To better understand these products, pure octinoxate was exposed to ozone and the PTR-TOF-MS spectra (after subtracting spectra without ozone exposure) is shown in S11 Fig of S1 File. Based on these spectra, the spectra from lotion as applied to skin and then exposed to ozone, and the structure of octinoxate, possible ozonation products are shown in S12 Fig of S1 File. There were increases in several prominent ions in group $C_8$: $C_8H_{11}O_2+$ '$C_8H_9O+$' $C_8H_9O_2+$. These could be (4-methoxyphenyl)methanol, its dehydration product and 4-methoxybenzaldehyde respectively. Yields of some $C_7$ ions (e.g. $C_7H_7O^+$) increased which may be the result of further fragmentation of octinoxate products. The notable negative yields in $C_7$ ions (Fig 4) are an artifact of decreasing emission rate, during the experiment, of volatile primary emissions from skin of

unknown compounds. The large increase in the $C_6$ group for lotion 5 is due in part to $C_6H_7O^+$, possibly phenol. It is not obvious how phenol might be formed from octinoxate, however, this ion was also observed during the pure octinoxate ozonation experiment. On skin, there was also an increase in nonanal and its dehydration product during ozonation of lotion 5, likely due to ozone reacting with the inactive ingredient oleth-3-phosphate.

Yield reductions could be due to physical changes in the lotions as well. Lotions contain substantial amounts of oily compounds such as fatty acid derivatives, polymers, humectants etc. This additional volume of material could slow the release of compounds simply by increasing the affinity of the products for the lipid/lotion solution (i.e. higher condensed-phase/gas-phase partition coefficient). The base lotion 1, as well as lotion 3, appear not to have a significant effect on yields of 6MHO, GA and decanal. The additives unique to lotion 2 are waxy lipids, which may react with ozone but also may alter the partitioning behavior. They can increase the viscosity of the film, thereby reducing the diffusivity of smaller molecules through the lipid/lotion mixture. Lotion 5 was noted by the researchers to be particularly viscous and oily. In addition to competition for ozone via reaction with octinoxate, which could reduce the ozonation rate of skin lipids, lotion 5 may also slow the release of products from the film to the gas phase.

### Dynamic emissions and physical changes in the skin surface film

Applying lotion to the skin alters the release dynamics of 6MHO, possibly pointing to changes in partitioning strength and viscosity. Shown in Fig 5 are the normalized, dynamic instrument signals, averaged over all participant results for the emissions of 6MHO. Signals are normalized to the average signal during the last 10 seconds of the ozone-on period so that all curves approach unity just prior to ozone being turned off. The signal associated with bare skin rises nearly, but not quite, to steady-state during the 300 s ozone-on period. For lotion 1, the signal

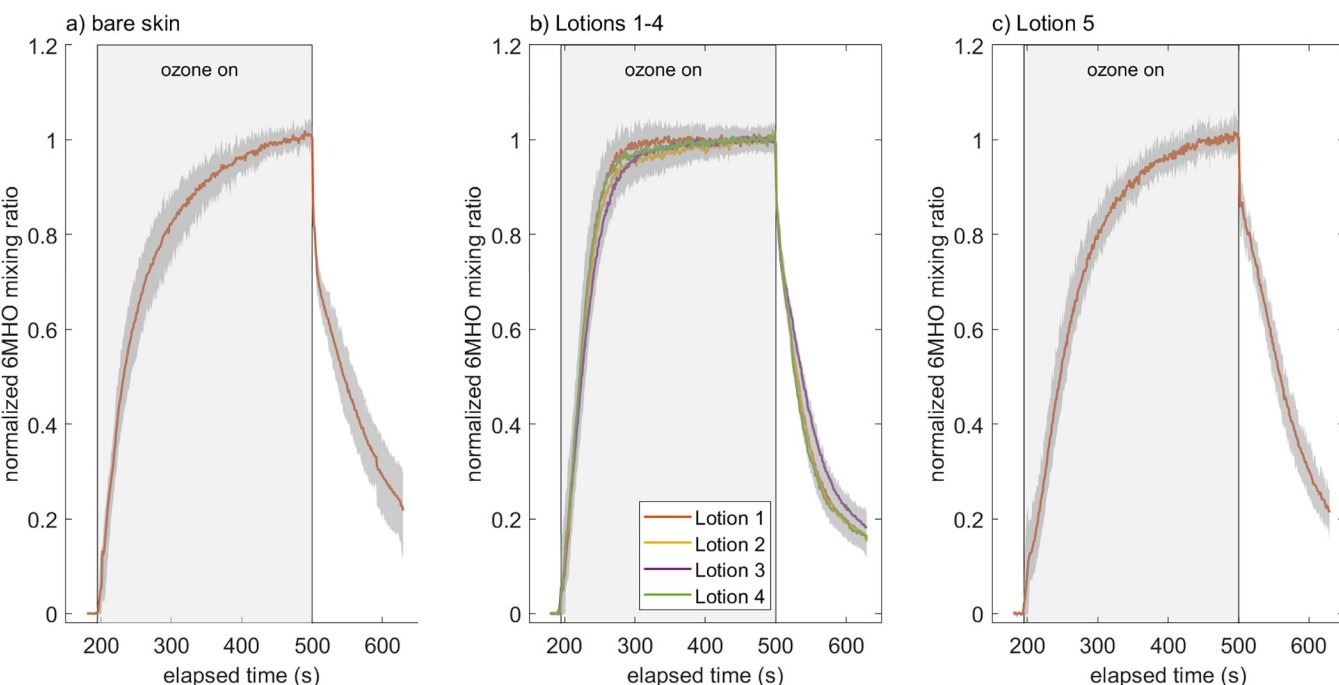

**Fig 5. Dynamic emissions of 6MHO.** Dynamic 6MHO signal before, during and after ozone exposure to skin, normalized by signal during the last 10 seconds of ozone exposure and averaged over all participant results. a) bare skin, b) Lotions 1–4, c) Lotion 5. Grey shaded square represents time period with ozone exposure. Darker grey shade around 6MHO curves represents 10th to 90th percentile results.

rises much more rapidly to an apparent steady-state in less than 100 s. We combine lotions 1–4 in one figure since the curves are very similar; this is consistent with the fact that lotions 2–4 are based on lotion 1. The dynamic rise in lotion 5 looks like that for bare skin. We hypothesize that these differences are primarily due to differences in partitioning strength.

We fit these dynamics to a double-exponential empirical model to obtain an estimate of the characteristic time to approach steady-state (see details in section S9 of S1 File). The primary outcome is that lotions 1 and 4 decrease the characteristic time, relative to bare skin, by about 36 seconds (from 59 to 23 seconds) even though the lotions substantially increase the total skin-surface layer thickness. This is likely due to dilution of skin lipids with hydrophilic compounds (such as propylene glycol) which results in a very large reduction in the 6MHO partition coefficient in the skin surface film (mixture of skin surface lipids and lotion compounds). The partition coefficient, $K_{SSF\text{-}g}$, is defined as the concentration in the skin surface film divided by the gas-phase concentration, at equilibrium. We estimated (see section S10 of S1 File) that applying lotion 1 to skin and allowing it to dry increases the thickness of the skin surface film by approximately factor of 5. By combining this dilution with a characteristic time analysis, we estimate that the partition coefficient, $K_{SSF\text{-}g}$, is reduced by over one order of magnitude for lotion 1. The characteristic times for lotions 2 and 3 are somewhat higher than that for lotions 1 and 4, suggesting that the waxy lipids and copolymer additives tend to decrease hydrophilicity and increase the 6MHO partition coefficient.

We did not see such clear evidence of differences in the dynamic release of another main product of squalene ozonation, geranyl acetone (see S14 Fig of S1 File). In our prior publication [20], we discuss that we observed unusual behavior suggesting that GA strongly sorbs to the walls of the sampling system, slowing its transmission to the Vocus PTR-MS. It is possible that sorption to tubing and other surface is responsible for most of the time-delay from source to detector and that skin partitioning delays are insignificant by comparison.

Lotion 1 (and probably in a similar fashion, lotions 2–4) appears to dilute skin lipids and reduce the 6MHO partition coefficient in the skin surface film. This has implications in not only the emission rate of reaction products from skin to air, but also the flux by diffusion through the stratum corneum. When a relatively hydrophobic 6MHO molecule is formed as a product within the skin surface film that includes lotion, it is present within a lipid-lotion mixture that is more hydrophilic than pure skin lipids. The air-to-film solubility of 6MHO is lower than it would be in pure skin surface lipids. For the same absolute concentration, the chemical activity (or fugacity) of 6MHO would then be *higher* in the mixture than in pure skin lipids. This means that the chemical driving force for transport through the stratum corneum would be higher with lotion present. It remains to be seen, by modeling [24, 25] or measurement, whether application of lotions would result in a higher net flux of 6MHO or other squalene ozonation products through the skin.

## Conclusions

Based on ozone uptake rates and product yields, we conclude that lotions alter the composition, properties, and chemistry of the skin surface film. We observe that lotions can reduce ozone uptake rates, both increase and decrease yields of ozone reaction products, reduce the partition coefficient of at least one important compound with the surface film and possibly reduce diffusivity of reactants and products in the film. Given the complexity of the system, more detailed research is warranted. This is especially so for companies hoping to design lotions that specifically protect skin from the harmful effects of reactive air pollution. Evidence that ozone promotes the formation of squalene hydroperoxides [26] suggests that lotions that lower ozone uptake or squalene oxidation rates could be protective. The methods described

here are relatively straightforward to apply in vivo and provide rapid gas-phase results. A skin-surface sampling and analysis method could be added that measures target condensed phase species, such as squalene hydroperoxides, to determine if gas-phase measurements are suitable for cosmetic product testing. Combined, these techniques can help test the antipollution claims of cosmetics [27, 28].

Lotions not only alter skin surface chemistry but could also influence indoor air quality. Ozone uptake to skin results in a wide array of volatile oxidized species and even the formation of submicron particles [9–16]. Some products of ozone-skin chemistry are themselves reactive and can significantly increase the ozone and hydroxyl radical reactivity of indoor air [13, 29, 30]. As we have shown, ozone also reacts with lotion ingredients generating volatile compounds unique to components in the lotions. By altering skin lipid oxidation and providing other chemical targets for ozonation, cosmetics could meaningfully impact indoor air chemistry.

## Supporting information

**S1 File.**
(DOCX)

## Acknowledgments

We thank Emer Duffy of Dublin City University for her gracious help and valuable discussions. This research was reviewed and approved by the University of North Carolina, Chapel Hill Institutional Research Board (IRB Study #18–3126).

## Author Contributions

**Conceptualization:** Glenn Morrison, Azin Eftekhari, Aixing Fan.

**Data curation:** Glenn Morrison, Azin Eftekhari, Francesca Majluf, Jordan E. Krechmer.

**Formal analysis:** Glenn Morrison, Azin Eftekhari, Francesca Majluf, Jordan E. Krechmer.

**Funding acquisition:** Glenn Morrison, Aixing Fan.

**Investigation:** Glenn Morrison, Azin Eftekhari, Aixing Fan, Francesca Majluf.

**Methodology:** Glenn Morrison, Azin Eftekhari, Aixing Fan, Francesca Majluf, Jordan E. Krechmer.

**Project administration:** Glenn Morrison.

**Resources:** Glenn Morrison, Aixing Fan, Jordan E. Krechmer.

**Supervision:** Glenn Morrison, Jordan E. Krechmer.

**Visualization:** Glenn Morrison, Jordan E. Krechmer.

**Writing – original draft:** Glenn Morrison, Azin Eftekhari, Francesca Majluf, Jordan E. Krechmer.

**Writing – review & editing:** Glenn Morrison, Azin Eftekhari, Aixing Fan, Jordan E. Krechmer.

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
