## [Decision Letter · Decision Letter 0]

7 Apr 2022

PONE-D-22-05746The influence of personal care products on ozone-skin surface chemistryPLOS ONE

Dear Dr. Morrison,

Thank you for submitting your manuscript to PLOS ONE. After careful consideration, we feel that it has merit but does not fully meet PLOS ONE’s publication criteria as it currently stands. Therefore, we invite you to submit a revised version of the manuscript that addresses the points raised during the review process.

We look forward to receiving your revised manuscript.

Kind regards,

Amitava Mukherjee, ME, Ph.D.

Academic Editor

PLOS ONE

Journal Requirements:

2. We noted in your submission details that a portion of your manuscript may have been presented or published elsewhere. [Some bare skin data was published in the following manuscript:

Morrison GC, Eftekhari A, Majluf F, Krechmer JE. Yields and Variability of Ozone Reaction Products from Human Skin. Environ Sci Technol. 2021 Jan 5;55(1):179–87.

Because this manuscript submitted to PLOS One focuses on the impact of personal care products on this chemistry, it is necessary to also include some bare-skin data as a comparison.] Please clarify whether this [conference proceeding or publication] was peer-reviewed and formally published. If this work was previously peer-reviewed and published, in the cover letter please provide the reason that this work does not constitute dual publication and should be included in the current manuscript.

Reviewers' comments:

Reviewer's Responses to Questions

**Comments to the Author**

1. Is the manuscript technically sound, and do the data support the conclusions?

Reviewer #1: Yes

Reviewer #2: Yes

2. Has the statistical analysis been performed appropriately and rigorously? 

Reviewer #1: Yes

Reviewer #2: Yes

3. Have the authors made all data underlying the findings in their manuscript fully available?

Reviewer #1: Yes

Reviewer #2: Yes

4. Is the manuscript presented in an intelligible fashion and written in standard English?

Reviewer #1: Yes

Reviewer #2: Yes

5. Review Comments to the Author

Reviewer #1: This is an interesting submission where authors explore the influence of skin lotions on the formation of product compounds through ozone reactions with human skin lipids. It was observed that lotions can decrease the ozone flux to the skin surface by 12 % to 25 %. Consequently, the yields of gas phase product compounds formed through ozonolysis of squalene were reduced in presence of commercial sunscreen, base lotion up to 70%.

I have some comments that may be of help for the authors to improve the content of the paper prior to publication.

1) It is ambiguous whether the lotions reduce the air pollution or increase the number of pollutants. Figure 3a shows that it is not obvious the influence of lotions on the yields of typical product compounds 6-MHO, GA and decanal formed through ozonolysis of squalene. Considering the uncertainties the yields are very similar from one to other case. On the other hand, Figure 4 shows that the lotions influence the yields of product compounds formed through ozone reactions with other components.

2) The authors should clarify better how are formed the compounds shown in Figure 4. What are the possible precursors? Is it only ozone involved or also secondarily formed OH radicals could contribute to the formation of these product compounds.

3) Some compounds which are ingredients of the lotions (e.g. 4-mehtoxybenzaldehyde, 4-methoxybenzoic acid) are known photosenstitizers which could enhance the heterogeneous reactions of ozone with the bare skin or skin coated with lotions and consequently affect the yields of the formed product compounds. For this reason it would be very interesting to test these experiments when sunlight irradiate the skin.

4) The manuscript contains only 14 citations which is unacceptable for this topic of research. It would be a great benefit for the readers to cite the relevant papers in this field starting from the papers related with the PTRMS, the discussed reaction mechanisms of ozonolysis of unsaturated compounds, the reactions of ozone with skin lipids, and so on and so forth.

5) It is not so clear for me how was attached the sampler to the skin avoiding the influence form the background compounds (e.g. acetone)

Minor:

I wonder how is derived Equation S3 for the deposition velocity of ozone. How was determined the uptake gamma (reaction probability of ozone) in this case?

Reviewer #2: Recheck the number of samples, as mentioned in line 52 (20 adults), while line 63 (21 volunteers).

The innovation of the manuscript is not reflected. Please add the innovation and research significance of the manuscript.

There is no section for conclusion. In addition to summarizing the actions taken and results, please strengthen the explanation of their significance and future work.

The findings of the current work needs more comparison with other researcher's findings / latest findings in the current area of research.

6. PLOS authors have the option to publish the peer review history of their article (what does this mean?). If published, this will include your full peer review and any attached files.

Reviewer #1: No

Reviewer #2: **Yes: **Che Zulzikrami Azner Abidin

---

## [Author Response · Author response to Decision Letter 0]

21 Apr 2022

Response to reviewer comments

We thank the reviewers for their thoughtful and thorough review comments. Below are responses to each comment with revised text quoted as appropriate.

Reviewer #1: This is an interesting submission where authors explore the influence of skin lotions on the formation of product compounds through ozone reactions with human skin lipids. It was observed that lotions can decrease the ozone flux to the skin surface by 12 % to 25 %. Consequently, the yields of gas phase product compounds formed through ozonolysis of squalene were reduced in presence of commercial sunscreen, base lotion up to 70%.

I have some comments that may be of help for the authors to improve the content of the paper prior to publication.

1) It is ambiguous whether the lotions reduce the air pollution or increase the number of pollutants. Figure 3a shows that it is not obvious the influence of lotions on the yields of typical product compounds 6-MHO, GA and decanal formed through ozonolysis of squalene. Considering the uncertainties the yields are very similar from one to other case. On the other hand, Figure 4 shows that the lotions influence the yields of product compounds formed through ozone reactions with other components.

Response: Yes, it is not so obvious just looking at figure 3a if there is a significant difference between bare skin and skin with lotion. We included symbols (*,o) that are referenced in the caption to indicate which yields were significantly different from bare skin (*) or lotion 1 (o). Also, with figure 3b we included the following, “Although not obvious from Figure 3a, yields from lotions 2, 4 and 5 are significantly lower than for bare skin. To make this easier to visualize, we divided the yield associated with each lotion and participant by the bare skin yields for that participant; this normalization, shown in Figure 3b, more explicitly shows the differences among lotions in a pair-wise fashion for each subject.” 

2) The authors should clarify better how are formed the compounds shown in Figure 4. What are the possible precursors? Is it only ozone involved or also secondarily formed OH radicals could contribute to the formation of these product compounds.

Response: unfortunately, we do not know what the precursors are or if OH may contribute. Based on the ingredient list and what is likely to be present, we discussed some hypothetical precursors: 

“Lotions 1-4, which are based on the same lotion (1) but with different additives, exhibit different yield patterns. Lotions 2-4 all exhibit increases in the yield of C7 ions dominated by C7H7O2+ and C7H5O+ which could represent benzoic acid and its dehydration product, respectively. Lotion 2 had distinctive increases in yields of groups C6 and C9. The additive ingredients of lotion 2, a proprietary mixture of waxy lipids could explain these groups, if the lipids themselves are unsaturated. There was a large relative increase in the C9 group for lotion 2 associated with nonenal (or an isomer) and its possible dehydration product C9H15+. Nonenal can be generated from the reaction of ozone with fats containing two or more double bonds such as linoleic acid or a similar. If linoleic acid is present, ozone attack at the omega-6 double-bond could also explain the increase in C6 alkyl fragments observed from lotion 2.

Lotion 5 had a very different pattern of yields from lotions 1-4. To better understand these products, pure octinoxate was exposed to ozone and the PTR-TOF-MS spectra (after subtracting spectra without ozone exposure) is shown in Figure S11. Based on these spectra, the spectra from lotion as applied to skin and then exposed to ozone, and the structure of octinoxate, possible ozonation products are shown in Figure S12. There were increases in several prominent ions in group C8: C8H11O2+ 'C8H9O+' C8H9O2+. These could be (4-methoxyphenyl)methanol, its dehydration product and 4-methoxybenzaldehyde respectively. Yields of some C7 ions (e.g. C7H7O+) increased which may be the result of further fragmentation of octinoxate products. The notable negative yields in C7 ions (Figure 4) are an artifact of decreasing emission rate, during the experiment, of volatile primary emissions from skin of unknown compounds. The large increase in the C6 group for lotion 5 is due in part to C6H7O+, possibly phenol. It is not obvious how phenol might be formed from octinoxate, however, this ion was also observed during the pure octinoxate ozonation experiment. On skin, there was also an increase in nonanal and its dehydration product during ozonation of lotion 5, likely due to ozone reacting with the inactive ingredient oleth-3-phosphate. “

3) Some compounds which are ingredients of the lotions (e.g. 4-mehtoxybenzaldehyde, 4-methoxybenzoic acid) are known photosenstitizers which could enhance the heterogeneous reactions of ozone with the bare skin or skin coated with lotions and consequently affect the yields of the formed product compounds. For this reason it would be very interesting to test these experiments when sunlight irradiate the skin.

Response: yes, we agree. Sunlight (especially UV) will certainly contribute to this chemistry.

4) The manuscript contains only 14 citations which is unacceptable for this topic of research. It would be a great benefit for the readers to cite the relevant papers in this field starting from the papers related with the PTRMS, the discussed reaction mechanisms of ozonolysis of unsaturated compounds, the reactions of ozone with skin lipids, and so on and so forth.

Response: We updated the paper with more citations to the relevant literature. 

5) It is not so clear for me how was attached the sampler to the skin avoiding the influence form the background compounds (e.g. acetone)

Response: The sampler is rather small and is well sealed to the skin with a thin silicone membrane. When operating, the cell is under a slight overpressure relative to the surrounding air, reducing the potential for air to be introduced. We altered the text as follows, “During operation, a 0.25mm thick gasket of medical grade silicone separated the surface sampler from the skin and created a seal. With air flowing, the samplers operate at a slight overpressure, preventing room air from entering the sampler.”

Minor:

I wonder how is derived Equation S3 for the deposition velocity of ozone. How was determined the uptake gamma (reaction probability of ozone) in this case?

Response: Equation S3 was derived by Cano-Ruiz et al. in their 1993 paper which is cited here. Note that we did not determine the reaction probability in this case. Instead, we assume that the highest deposition velocity observed is due to a “high” reaction probability and use this as the transport limited deposition velocity in Equation S3. Rather than choose a specific reaction probability, we instead plot Equation S3 for a range of reaction probabilities in Figure S10. This allows us to argue that the ozone deposition velocity can remain high even after skin lipids have been substantially diluted with lotions.

Reviewer #2: Recheck the number of samples, as mentioned in line 52 (20 adults), while line 63 (21 volunteers).

Response: yes, this could be unclear. We had 21 volunteers, but had to exclude data from one individual (incomplete data collection). We edited this in the methods section, “A convenience sample of 21 volunteers participated in this study; because of incomplete data collection, results from one participant are excluded from the analysis. Therefore, we include results for a 20 participants(17 female, 3 male; age 20-61 y).”

The innovation of the manuscript is not reflected. Please add the innovation and research significance of the manuscript.

Response: we added the following at the end of the introduction:, “To our knowledge, this research is the first to assess the influence of lotions on ozone-skin lipid chemistry, which has implications for both skin health and indoor air quality.”

There is no section for conclusion. In addition to summarizing the actions taken and results, please strengthen the explanation of their significance and future work.

Response: The last paragraph served as our conclusions section, but we have separated this out with the “Conclusions” heading. We also edited it to address this comment: “Based on ozone uptake rates and product yields, we conclude that lotions alter the composition, properties, and chemistry of the skin surface film. We observe that lotions can reduce ozone uptake rates, both increase and decrease yields of ozone reaction products, reduce the partition coefficient of at least one important compound with the surface film and possibly reduce diffusivity of reactants and products in the film. Given the complexity of the system, more detailed research is warranted. This is especially so for companies hoping to design lotions that specifically protect skin from the harmful effects of reactive air pollution. Evidence that ozone promotes the formation of squalene hydroperoxides(26) suggests that lotions that lower ozone uptake or squalene oxidation rates could be protective. The methods described here are relatively straightforward to apply in vivo and provide rapid gas-phase results. A skin-surface sampling and analysis method could be added that measures target condensed phase species, such as squalene hydroperoxides, to determine if gas-phase measurements are suitable for cosmetic product testing. Combined, these techniques can help test the antipollution claims of cosmetics(27,28). 

Lotions not only alter skin surface chemistry but could also influence indoor air quality. Ozone uptake to skin results in a wide array of volatile oxidized species and even the formation of submicron particles (9–16). Some products of ozone-skin chemistry are themselves reactive and can significantly increase the ozone and hydroxyl radical reactivity of indoor air(13,29,30). As we have shown, ozone also reacts with lotion ingredients generating volatile compounds unique to components in the lotions. By altering skin lipid oxidation and providing other chemical targets for ozonation, cosmetics could meaningfully impact indoor air chemistry.”

The findings of the current work needs more comparison with other researcher's findings / latest findings in the current area of research.

Response: Where possible, we compared our research to others and have now cited much more of the relevant literature. Bare-skin yield comparisons with the (few) other papers out there were made in our previous publication. To our knowledge, there have been no reports of the influence of lotion on skin lipid ozonation chemistry for comparison. There have been some reports on the impact of “antioxidant” lotions, but their results are not directly comparable. We acknowledge that literature, but do not attempt to make direct comparisons.

---

## [Decision Letter · Decision Letter 1]

26 Apr 2022

The influence of personal care products on ozone-skin surface chemistry

PONE-D-22-05746R1

Dear Dr. Morrison,

We’re pleased to inform you that your manuscript has been judged scientifically suitable for publication and will be formally accepted for publication once it meets all outstanding technical requirements.

Kind regards,

Amitava Mukherjee, ME, Ph.D.

Academic Editor

PLOS ONE

Additional Editor Comments (optional):

Reviewers' comments:

Reviewer's Responses to Questions

**Comments to the Author**

1. If the authors have adequately addressed your comments raised in a previous round of review and you feel that this manuscript is now acceptable for publication, you may indicate that here to bypass the “Comments to the Author” section, enter your conflict of interest statement in the “Confidential to Editor” section, and submit your "Accept" recommendation.

Reviewer #1: All comments have been addressed

Reviewer #2: All comments have been addressed

2. Is the manuscript technically sound, and do the data support the conclusions?

Reviewer #1: Yes

Reviewer #2: Yes

3. Has the statistical analysis been performed appropriately and rigorously? 

Reviewer #1: Yes

Reviewer #2: Yes

4. Have the authors made all data underlying the findings in their manuscript fully available?

Reviewer #1: Yes

Reviewer #2: Yes

5. Is the manuscript presented in an intelligible fashion and written in standard English?

Reviewer #1: Yes

Reviewer #2: Yes

6. Review Comments to the Author

Reviewer #1: The authors have properly addressed my comments and the manuscript can be published in the current form.

Reviewer #2: The manuscript has been revised accordingly, and has been enhanced compared with the previous submission.

7. PLOS authors have the option to publish the peer review history of their article (what does this mean?). If published, this will include your full peer review and any attached files.

Reviewer #1: No

Reviewer #2: **Yes: **Che Zulzikrami Azner Abidin

---

## [Editor Report · Acceptance letter]

11 May 2022

PONE-D-22-05746R1 

The influence of personal care products on ozone-skin surface chemistry 

Dear Dr. Morrison:

I'm pleased to inform you that your manuscript has been deemed suitable for publication in PLOS ONE. Congratulations! Your manuscript is now with our production department. 

Kind regards, 

on behalf of

Professor Dr. Amitava Mukherjee 

Academic Editor

PLOS ONE